# Parental teasing and body dissatisfaction in White and South Asian females: An exploratory cross-sectional analysis using moderated mediation

**Amané Halicki-Asakawa**●, **Dilpreet Lidder**●, **Maya Libben**●*

Department of Psychology, University of British Columbia Okanagan, Kelowna, British Columbia, Canada

* maya.libben@ubc.ca

## Abstract

Parental weight-related teasing can be associated with body dissatisfaction among children and adolescents. Though this relationship is frequently observed among White females, limited research has been conducted with other ethnic groups. Previous research has demonstrated a strong link between familial factors and psychosocial functioning in South Asian females. Therefore, it may be the case that similar associations between parental weight teasing and body dissatisfaction can be found in this population. A sample of 352 young adult females, comprising 58.7% White and 42.3% South Asian, all Canadian residents, participated in this exploratory, cross-sectional study. They completed the Perception of Teasing Scale, the Body Shape Questionnaire, the Rosenberg Self-Esteem Scale, and the Beck Depression Inventory. South Asian participants reported higher rates of parental weight-related teasing compared to White participants. For both groups, the association between parental weight teasing and body dissatisfaction showed an indirect path through depressive symptoms, with higher teasing associated with greater depressive symptoms and concurrent body dissatisfaction. The association between teasing and depressive symptoms was moderated by self-esteem, showing a stronger relationship for those with high self-esteem (White) or moderate to high self-esteem (South Asian). Additionally, self-esteem was directly associated with body dissatisfaction among White participants but not in South Asian participants. These exploratory findings provide preliminary evidence of a link between parental teasing and body dissatisfaction in South Asian females, emphasizing the need to consider familial and cultural factors in body dissatisfaction prevention efforts. The importance of ethnic-specific considerations for body dissatisfaction interventions are highlighted, while recognizing that additional longitudinal research is needed to further elucidate this relationship.

License, which permits unrestricted use, distribution, and reproduction in any medium, provided the original author and source are credited.

**Data availability statement:** Data is available at https://osf.io/vahrz/?view_only=1b3f6f4c-9f8a4216927ac30d06345714.

**Funding:** The authors received no specific funding for this work.

**Competing interests:** The authors have declared that no competing interests exist.

## Introduction

Body dissatisfaction (BD) describes an individual's subjective negative evaluation of their weight, body shape, and size (Garner, 2002) [1] . BD has been linked to numerous adverse physical and mental health correlates, such as low self-esteem, obesity, depressive symptoms, stress, social anxiety, and impaired sexual functioning (Cash, 2012) [2]. Additionally, BD is a recognized risk factor for the development and maintenance of eating disorders, including anorexia nervosa, bulimia nervosa, and binge eating disorder (Cooley & Toray, 2001) [3]. While BD is known to occur across genders, prevalence rates are significantly higher for those identifying as female (e.g., [4]). Concerningly, the age at which females report experiencing BD has consistently decreased over recent decades (Damiano et al., 2015) [5]. This trend underscores the importance of examining potential psychosocial and developmental correlates of BD, which include biological, psychological, and socio-developmental factors such as parental influence.

Parents and caregivers play a formative role in the developmental course of body image during childhood. As key socio-environmental role models, their attitudes and behaviors regarding body image, diet, and exercise are often internalized by children (e.g., [6], Wansink et al., 2017) [7]. For example, discouraging certain eating behaviors based on beliefs surrounding obesity, health issues, and appearance can influence body image and BD development in children [8]. Furthermore, negative comments including teasing have been identified as particularly harmful parental practices, especially when directed toward the child's weight and/or physical appearance. Teasing involves verbal taunts that serve as a form of provocation, mockery or criticism, often balanced with playful intonations or gestures to soften the impact [9].

Parental weight teasing has been associated with a myriad of negative psychological outcomes (see Dahill et al., 2021 for a review) [10] and is considered a risk factor for BD [11]. For example, using the Perceptions of Teasing Scale (POTS; Thompson et al., 1995) [12], Keery and colleagues [13] found that 23% of a sample of 372 middle school girls reported appearance-related teasing by a parent. When controlling for Body Mass Index (BMI), the authors found that parental teasing predicted BD, social comparison, thin-ideal internalization, dietary restriction, bulimic behaviors, lower self-esteem, and increased depressive symptoms. Similarly, Pötzsch and colleagues [11] found that perceived parental weight-related teasing among adolescents significantly predicted global eating disorder psychopathology.

Additional evidence links teasing to broader psychological distress. Eisenberg and colleagues [14] found that adolescents who reported frequent weight-based teasing from family members were more likely to report depressive symptoms, poor self-esteem, and suicidal ideation, independent of BMI and other demographic variables. In a longitudinal study, Haines and colleagues [15] followed adolescents over five years and reported that weight-related teasing from family members predicted increased BD, dieting behaviors, and clinically significant depressive symptoms at follow-up. Neumark-Sztainer and colleagues [16] similarly found that both male and female adolescents who experienced weight-teasing by family members exhibited lower self-esteem and were more likely to engage in unhealthy weight control

practices, including extreme dieting and binge eating. In a slightly older college-aged sample, Quick and colleagues [17] demonstrated that self-esteem acted as a partial mediator in the relationship between weight teasing and disordered eating behaviors. The authors suggest that internalized negative self-perceptions stemming from teasing experiences may play a central role in the emergence of eating pathology and BD. Finally, in a one-year longitudinal study Webb and colleagues [18] demonstrated that parental weight-related teasing in early adolescence prospectively predicted increases in emotional eating one year later, highlighting the longitudinal impact of early teasing on the development of disordered eating behaviors.

Historically, much of the literature on parental weight teasing and BD has focused on White samples (e.g., [13]), due in part to outdated notions that eating disorders are culture-bound to North American females, and the persistent under-representation of non-White individuals in health research [19]. However, there is now evidence that non-White populations, including South Asian females, experience similar risk factors and vulnerabilities for eating pathology and BD (e.g., [20]). Rates of eating disorders and BD observed in South Asian females are comparable, or higher, to those observed in White females [21,22]. Additionally, BD and self-awareness of body size have been found to emerge at younger ages for this population, which may be due in part to comments regarding body size from family members (pallan et al., 2011) [20,23].

South Asian minorities living in Westernized countries may be at particular risk for BD given the presence of other psychological factors tied to acculturation, racism and geographical location (e.g., Abbas et al., 2010) [24]. Previous research on minority South Asian women in the UK and Canada found greater levels of BD and maladaptive eating attitudes compared to White women (e.g., Kennedy et al., 2004) [25]. Furthermore, qualitative themes of sheltered upbringings, parental over-protection/control, and discouragement from assimilation into Western culture have emerged as potential contributors to disordered eating for South Asian women in Canadian contexts [26]. Certain adverse effects of familial practices on other areas of psycho-social functioning have also been noted for minority South Asian females (e.g., emotion regulation; [27]).

Given the influential role of family in many South Asian communities, especially where cultural and generational expectations may conflict with Western norms, it is important to consider whether models linking parental teasing and BD developed in predominantly White samples apply equally to this group. While parental teasing is associated with poorer body image across cultures, the meaning and psychological impact of teasing may be shaped by values such as familial honor, collectivism, and gendered appearance norms, making direct generalizations problematic. One of the few studies in this area, by Reddy and Crowther [28], found preliminary evidence of a link between weight- and shape-related teasing and BD among American South Asian females, but did not focus specifically on parental teasing or test possible mediating or moderating variables.

The current study examined the cross-sectional association between parental teasing and BD among White and South Asian young adult females living in Canada. Building on prior findings suggesting that parental weight teasing may contribute to negative body image, this study adopted an exploratory approach, recognizing that variables such as depressive symptoms and self-esteem may serve different roles (e.g., mediators, moderators, or confounders) depending on the analytic model employed (Singh-Manoux, 2005) [29]. As such, we aimed to address the following research questions using a moderated mediation framework: (1) Do self-reported levels of parental weight related teasing differ between White and South Asian participants? (2) Is parental teasing associated with BD, and is this relationship mediated by depressive symptoms? and (3) Does self-esteem moderate associations between teasing, depressive symptoms, and BD, consistent with a moderated mediation framework? This research aims to contribute to an initial understanding of the socio-developmental context surrounding BD, highlighting parental weight-related teasing as a culturally relevant correlate in both White and South Asian communities. It also seeks to emphasize the necessity of culturally sensitive interventions and preventive strategies, particularly for populations in which familial influences carry significant emotional weight. By examining BD among South Asian and White women, this study aims to inform approaches that are responsive to culturally specific pathways linking parental behavior and body image concerns, while acknowledging that future longitudinal studies will be required to formally test directional and causal mechanisms.

## Method

### Participants

Participants were 352 female undergraduate students (*M*age = 19.76, SD = 1.38) recruited from psychology courses at a Canadian university offering research participation credit. Eligibility criteria included identifying as either White or South Asian, being between the ages of 18 and 25, and being able to provide informed consent. For the South Asian group, inclusion required either being born in a South Asian country or having at least one parent or guardian of South Asian origin. All participants were required to have sufficient English proficiency to understand the study instructions and access to a computer, phone or tablet to complete the study online.

Ethics approval was obtained from the Behavioural Research Ethics Board at the University of British Columbia Okanagan (H21-03408), and data collection occurred between March 12, 2021, and July 14, 2022. Participants completed the study remotely and received course credit for their participation. Informed consent and assurances of anonymity were provided at the start of the study.

203 participants (57.7%) self-identified as White and 149 (42.3%) as South Asian. Body mass index (BMI), calculated as weight in kilograms divided by height in meters squared, ranged from 15.42 to 37.41 (*M*BMI = 22.7, SD = 3.7). There were no statistically differences in BMI between the two ethnic groups (White: *M*BMI = 22.8; South Asian: *M*BMI = 22.6; p >.05).

### Procedure

The current study was administered online via the Qualtrics survey platform (Qualtrics, Provo, UT). Upon completing a digital consent form, participants were presented with a demographics questionnaire that included items pertaining to ethnic identity, immigration status, parental and familial characteristics, age, height and weight. Subsequently, participants completed the Body Shape Questionnaire (Cooper et al., 1987) [30], Rosenberg Self-Esteem Scale (Rosenberg, 1965) [31], Perceptions of Teasing Scale [32], and Beck Depression Inventory-II (Beck et al., 1996; Beck et al., 1961) [33,34] counterbalanced for order. The study took approximately 30-min to complete. No *a priori* registration of the analysis plan was conducted, which aligns with the exploratory nature of this research.

### Materials

**Parental teasing.** The Perception of Teasing Scale (POTS; [32]) was used as a measure of parental teasing. The POTS is an 11-item scale measuring the respondent's experiences of teasing surrounding weight (e.g., "jokes about you being heavy") and competency (e.g., "said you acted dumb"), as well as the level of distress in response to the teasing experience on a 5-point scale with higher ratings indicating more distress. Initially, weight teasing (i.e., directed towards the individual's body size) and competence teasing (i.e., directed towards non-weight related abilities) were considered; however, competence teasing was ultimately excluded from focal analyses as it was not hypothesized to strongly relate to BD. For the purpose of the current study, the teaser was specified to be the respondent's parent or guardian and the period of teasing was specified to occur between the ages of 5–16 years. The POTS has been demonstrated to be a valid and reliable measure of teasing [35].

**Body dissatisfaction.** The Body Shape Questionnaire (BSQ; Cooper et al., 1987) is a 34-item scale measuring the respondent's BD and perceptions concerning their body shape over the past four weeks using a 6-point Likert rating scale ranging from 1 (i.e., *never*) to 6 (i.e., *always*), with higher scores indicating higher BD. The BSQ has been validated and used in various clinical and research settings [36].

**Self esteem.** The Rosenberg Self-Esteem Scale (RSE; Rosenberg, 1965) is a 10-item scale that assesses the respondent's positive and negative feelings about themselves. Each item is rated on a 4-point Likert scale ranging from 1 (strongly disagree) to 4 (strongly agree). The scale includes both positively worded items (e.g., "I feel that I have a number

of good qualities") and negatively worded items (e.g., "At times I think I am no good at all"), with the latter requiring reverse scoring. After reverse coding the negatively worded items, all item scores are summed to create a total score, with higher scores indicating higher global self-esteem. The RSE has been validated and widely used in various clinical and research settings [37].

**Depressive symptoms.** The Beck Depression Inventory-II (BDI-II; Beck et al., 1996) is a 21-item scale that assesses depressive symptoms. Each item is scored using a 4-point scale ranging from 0 (i.e., absence of the symptom) to 3 (i.e., severe presence of the symptom). A total score is calculated, with higher scores indicating the severity of depressive symptoms The BDI is a validated and widely used measure of depressive symptoms in both clinical and research settings [38].

## Variables

To address the research questions outlined above, the following quantitative variables were defined. Parental weight teasing was the primary exposure (predictor) variable, assessed through the POTS. BD, measured by the BSQ, was the primary outcome variable. Depressive symptoms, assessed via the BDI-II, were examined as a potential mediator of the relationship between parental teasing and BD. Self-esteem, assessed with the RSE was investigated as a potential effect modifier (moderator) of the relationship between parental teasing and depressive symptoms. Participant ethnicity (White or South Asian) was treated as a subgroup to explore potential cultural differences in these relationships. BMI, calculated using self-reported height and weight ($kg/m^2$), was considered a potential covariate.

All variables in the study were analyzed as continuous variables. For analyses involving moderation, self-esteem scores were mean centered prior to interaction analyses to reduce potential multicollinearity and improve interpretability of interaction effects. BMI was also treated continuously, given its distribution and clinical relevance, and included cautiously as a covariate rather than as a mediator. No additional groupings of continuous variables were performed, aside from subgroup analysis based on participant ethnicity, which was categorically defined.

## Statistical approach

Statistical analyses were conducted using IBM SPSS version 29. Initial analyses included descriptive statistics, means comparisons (t-tests), and Pearson correlation analyses. Regression assumptions (linearity, homoscedasticity, normality of residuals, and collinearity diagnostics) were tested and confirmed prior to mediation and moderated mediation analyses. Mediation analyses were conducted separately within White and South Asian subgroups to examine the relationships between parental weight bias, self-esteem, and BD using Model 4 of the PROCESS macro, a logistic regression path analysis tool, for SPSS version 29 [39]. A bootstrap resampling process was repeated 5,000 times to generate a 95% confidence interval (CI), with indirect effects considered significant if the CI did not cross zero [39]. Moderated mediation analyses were subsequently carried out using Model 8 of the PROCESS macro [39] to test whether self-esteem moderated the associations between weight teasing, depressive symptoms, and BD across both groups. Simple slopes were examined at high (+1 SD), moderate (mean), and low (–1 SD) levels of self-esteem. The potential confounding effect of BMI was statistically adjusted for by including it as a covariate in the mediation and moderated mediation models; however, caution was exercised regarding its inclusion in analytical models to avoid introducing potential bias. Missing data were minimal (less than 5% per variable) and handled using listwise deletion. Because the sample was recruited using convenience sampling from an undergraduate psychology course, no special adjustments for sampling strategy were necessary.

## Results

Immigrant status of participants and their caregivers is presented in Table 1 along with primary caregiver information. White participants were statistically more likely to have been raised by single parents/caregivers or both parents/caregivers while South Asian participants were more likely to have been raised by multiple caregivers. South Asian participants were more likely to be immigrants to Canada and to have been raised by immigrant caregivers.

**Table 1. Self-reported family characteristics.**

| | n Endorsed and % of Group | | |
| | White (n = 203) | South Asian (n = 149) | p |
|---|---|---|---|
| Childhood caregiver(s) | | | |
| Single-parent | 15 (7.4%) | 9 (6.0%) | <.001 |
| Both parents | 178 (87.7%) | 121 (81.2%) | <.001 |
| Multiple caregivers | 10 (4.9%) | 19 (12.8%) | <.001 |
| Caregiver immigrant status | | | |
| Immigrant(s) | 35 (17.2%) | 140 (94.0%) | <.001 |
| Non-immigrant(s) | 168 (82.8%) | 9 (6.0%) | <.001 |
| Participant immigrant status | | | |
| Immigrant | 11 (5.4%) | 91 (61.1%) | <.001 |
| Non-immigrant | 192 (94.6%) | 58 (38.9%) | <.001 |

*Note.* Table presents self-reported caregiver structure and immigration status for participants and their childhood caregivers, separately by ethnocultural group. "Caregiver immigrant status" indicates whether one or both primary caregivers were born outside of Canada. "Participant immigrant status" indicates whether the participant themselves was born outside of Canada.

p-values reflect results of t-tests comparing frequencies between groups.

Mean results from self-report measures and internal consistency across both White and South Asian groups are summarized in Table 2. Overall, South Asian participants reported statistically higher rates of parental weight bias on the POTS weight teasing subscale (*M* = 1.85, *SE* = 1.1) compared to White participants (*M* = 1.29, *SE* = 0.48; *t*(350) = −7.16, *p* =.001, *d* = −0.77), though both groups reported similar rates of distress due to teasing (*t*(271) = −1.99, *p* =.05, *d* = −0.24). There were no statistical differences in self-reported BD (BSQ), depressive symptoms (BDI-II) or self-esteem (RSE) between groups. Correlation analyses indicated significant relationships between all self-report variables (i.e., weight bias, BD, self-esteem, and depressive symptoms) for both White and South Asian participants (Table 3). Internal consistency was acceptable to excellent for all measures across both groups.

**Table 2. Comparison of mean scores on self-reported variables.**

| Measure | White | | South Asian | | |
| | M (SD) | α | M (SD) | α | t |
|---|---|---|---|---|---|
| Body Mass Index | 22.8 (3.7) | | 22.6 (3.7) | | 0.537 |
| Weight Teasing[a] | 1.3 (0.5) | .81 | 1.9 (1.0) | .91 | −7.158*** |
| Body Dissatisfaction[b] | 103.8 (42.1) | .98 | 105.3 (41.2) | .91 | −0.339 |
| Self-Esteem[c] | 18.2 (6.1) | .91 | 18.6 (5.9) | .90 | −0.656 |
| Depressive Symptoms[d] | 15.0 (11.4) | .93 | 15.5 (10.6) | .91 | −0.468 |

*Note.* Comparison of Mean Scores and Standard Deviations for White (n = 203) and South Asian (n = 149) groups on self-report measures. Cronbach's alpha (α) values reflect the internal consistency reliability of each scale within each group.

[a]Perceptions of Teasing Scale, Weight Teasing Subscale;

[b]Body Shape Questionnaire;

[c]Rosenberg Self-Esteem Scale;

[d]Beck Depression Inventory-II.

*p* <.05, **p* <.01, ***p* <.001.

**Table 3. Combined correlations for White (above the diagonal) and South Asian (below the diagonal) participants.**

|  | BSQ[a] | BMI[b] | SEQ[c] | BDI[d] | POTS-W[e] |
|---|---|---|---|---|---|
| BSQ[a] | 1.00 (W) | .33*** (W) | .59*** (W) | .58*** (W) | .44*** (W) |
| BMI[b] | .43*** (SA) | 1.00 (W) | .01 (W) | −.02 (W) | .45*** (W) |
| SEQ[c] | .38*** (SA) | .10 (SA) | 1.00 (W) | .81*** (W) | .28*** (W) |
| BDI[d] | .49*** (SA) | .13 (SA) | .78*** (SA) | 1.00 (W) | .27*** (W) |
| POTS-W[e] | .61*** (SA) | .50*** (SA) | .36*** (SA) | .42*** (SA) | 1.00 (SA) |

*Note.* Pearson correlation coefficients between self-report variables for White (n = 203; W) and South Asian (n = 149; SA; below diagonal in unshaded cells) participants.

[a]Body Shape Questionnaire;

[b]Body Mass Index;

[c]Rosenberg Self-Esteem Scale;

[d]Beck Depression Inventory-II;

[e]Perceptions of Teasing Scale, Weight Teasing Subscale.

*p <.05, **p <.01, ***p <.001.

Before proceeding with our primary analyses, we ensured that our data met all assumptions required for regression-based analyses [40]. A visual inspection of the data indicated that the assumptions of linearity, homoscedasticity, and normality were met for both samples. The data met the assumptions of independent errors in both the White (*Durbin-Watson value* = 2.06) and South Asian samples (*Durbin-Watson value* = 1.73). Additionally, there was no evidence of multicollinearity in either the White (BMI, *VIF* = 1.28; depressive symptoms, *VIF* = 2.87; self-esteem, *VIF* = 2.87; weight teasing, *VIF* = 1.40) or South Asian groups (BMI, *VIF* = 1.35; depressive symptoms, *VIF* = 2.73; self-esteem, *VIF* = 2.58; weight teasing, *VIF* = 1.61).

## Tests of mediation

Associations among parental weight teasing, depressive symptoms, and BD were examined separately for White and South Asian participants. In both groups, greater parental weight teasing was associated with higher levels of depressive symptoms, which in turn were positively associated with greater BD. These findings support an indirect pathway, wherein depressive symptoms partially mediate the relationship between parental teasing and BD.

For the White sample, the direct effect of parental teasing on BD remained statistically significant (*B = 3.07, SE = 0.64, p <.001, 95% CI [1.80, 4.34]*), and the indirect effect through depressive symptoms was also statistically supported (*B = 0.11, SE = 0.05, 95% CI [0.03, 0.22]*). This suggests that parental teasing contributes to BD both directly and indirectly via depressive symptoms.

Similarly, for the South Asian sample, the direct association between parental teasing and BD was statistically significant (*B = 3.27, SE = 0.90, p <.001, 95% CI [1.49, 5.01]*), and the indirect effect through depressive symptoms was also supported (*B = 0.10, SE = 0.04, 95% CI [0.03, 0.19]*). Thus, the mediation pathway was consistent across both groups, highlighting depressive symptoms as a potential mechanism linking parental weight teasing to BD.

## Tests of moderated mediation

To explore whether self-esteem moderated these mediation pathways, moderation analyses were conducted separately for each group. Self-esteem was specifically tested as a moderator of the relationship between parental weight teasing and depressive symptoms. Results of the moderated mediation analysis for both samples, as well as the effect sizes and confidence intervals for these associations, can be found in Table 4.

**White participants.** Moderation analyses revealed that self-esteem altered the relationship between parental weight teasing and depressive symptoms (*B = 0.47, t(198) = 2.37, p =.019*). To clarify the nature of the interaction, participants

**Table 4. Evaluation of self-esteem as a moderator of predictor-mediator and predictor-criterion associations.**

| Predictor Variable | White Sample | | | | | | South Asian Sample | | | | | |
|---|---|---|---|---|---|---|---|---|---|---|---|---|
| | b | t | p | R2 | F | df | b | t | p | R2 | F | df |
| Criterion: Depressive Symptoms[a] | | | | | | | | | | | | |
| Model Summary | | | <.001 | .66 | 96.77 | (4, 198) | | | <.001 | .65 | 67.42 | (4, 144) |
| Weight Teasing[b] | .47 | .36 | .084 | | | | .70 | 1.95 | .072 | | | |
| Self-Esteem[c] | .08 | 17.99 | <.001 | | | | 1.33 | 13.83 | <.001 | | | |
| Weight Teasing[b] x Self-Esteem[c] | .36 | 2.37 | .019 | | | | .31 | 2.72 | .007 | | | |
| Body Mass Index | −.17 | .14 | .227 | | | | −.07 | −.41 | .685 | | | |
| Criterion: Body Dissatisfaction[d] | | | | | | | | | | | | |
| Model Summary | | | <.001 | .51 | 40.63 | (5, 197) | | | <.001 | .47 | 25.49 | (5, 143) |
| Weight Teasing[b] | 5.73 | 2.69 | .097 | | | | 17.33 | 5.08 | .023 | | | |
| Self-Esteem[c] | 2.06 | 3.52 | .005 | | | | −.47 | −.66 | .667 | | | |
| Weight Teasing[b] x Self-Esteem[c] | −.61 | −.88 | .379 | | | | −.51 | −.92 | .362 | | | |
| Depressive Symptoms[a] | 1.12 | .32 | <.001 | | | | 1.39 | 3.47 | <.001 | | | |
| Body Mass Index | 3.07 | 4.77 | <.001 | | | | 2.19 | 2.76 | .007 | | | |

Note. β = standardized regression coefficient; $R^2$ = coefficient of determination; df = degrees of freedom. Interaction terms are the product of centered predictor and moderator variables.

[a]Beck Depression Inventory-II;

[b]Perceptions of Teasing Scale, Weight Teasing Subscale;

[c]Rosenberg Self-Esteem Scale;

[d]Body Shape Questionnaire.

were grouped by self-esteem levels: high (i.e., 1 *SD* above the mean; n = 29), moderate (i.e., mean-level; n = 140), and low (i.e., 1 *SD* below the mean; n = 34). The strongest association between teasing and depressive symptoms occurred for participants with high self-esteem (*B = 2.79, t(198) = 2.20, p =.029*), indicating that among White females with high self-esteem, greater parental teasing was associated with higher levels of depressive symptoms. For those with moderate (*B = 0.47, t(198) =.36, p =.718*) or low self-esteem (*B = −1.77, t(198) = −.94, p =.350*), the relationship between teasing and depressive symptoms was weaker and not statistically supported (see Fig 1).

Parental weight teasing showed the most substantial relationship with depressive symptoms in White females with high self-esteem (approximately 17.7% of the sample). Moreover, the indirect effect of teasing on body dissatisfaction through depressive symptoms was statistically significant only for participants with high self-esteem, suggesting that the mediation pathway was supported in this group but not in those with moderate or low self-esteem. This indicates a partial mediation effect among high self-esteem White participants, where teasing is linked to BD both directly and indirectly via depressive symptoms.

Additionally, depressive symptoms were positively associated with BD, independently of self-esteem. BMI was also positively associated with BD (*B = 3.07, F(197) = 40.53, p =.0001*). However, self-esteem did not moderate the direct association between parental teasing and BD itself (*B = −.61, t(197) = −.88, p =.379*), suggesting that teasing was associated with BD to a similar extent regardless of self-esteem levels.

**South Asian participants.** Similarly, for South Asian participants, self-esteem moderated the relationship between parental weight teasing and depressive symptoms (*B = 0.70, t(144) = 2.72, p =.007*). Using the same procedure as described above for the White sample, splitting South Asian participants by levels of self-esteem (high n = 22; moderate, n = 101; low, n = 26) suggested that the strongest positive associations between teasing and depressive symptoms occurred for those with moderate (*B = 3.17, t(144) = 3.91, p =.000*) and high self-esteem (*B = 1.37, t(144) = 1.95, p =.053*). No clear associations were found for South Asian participants with low levels of self-esteem (*B = −0.43, t(144) = −.39, p =.684*).

*Moderated Mediation Models of Teasing, Depressive Symptoms, and Body Dissatisfaction by Group and Self-Esteem*

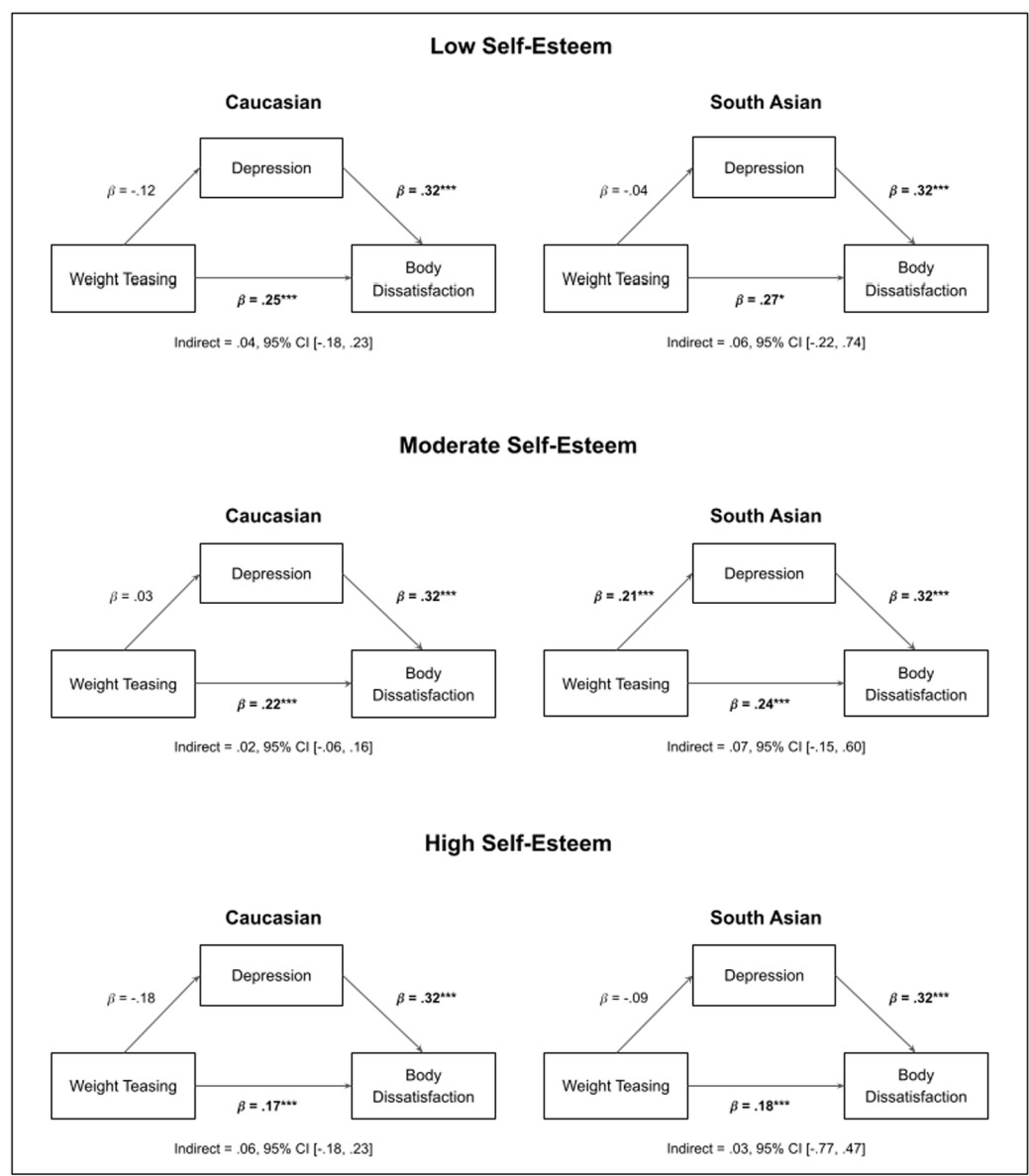

*Note.* This figure presents moderated mediation models examining the relationship between parental weight teasing and body dissatisfaction, mediated by depression and moderated by self-esteem. Models are grouped by ethnicity (Caucasian vs. South Asian) and split by self-esteem level (low, moderate, high). Standardized beta coefficients are displayed for each path, with significance indicated by asterisks (*p < .05, **p < .01, ***p < .001). Indirect effects and corresponding 95% confidence intervals are reported below each model. Depression-to-body dissatisfaction paths are held constant across self-esteem levels for consistency with PROCESS Model 8 output. BMI was included as a covariate but is not depicted to enhance figure clarity.

**Fig 1. Moderated mediation models of teasing, depressive symptoms, and body dissatisfaction by group and self-esteem.**

As such, the relationship between parental teasing and depressive symptoms was most pronounced for South Asian participants reporting either moderate or high self-esteem, representing a majority (52.4%) of the sample (see Fig 1). In addition, the indirect effect of teasing on BD through depressive symptoms was statistically significant for South Asian participants with moderate and high self-esteem, but not for those with low self-esteem. This pattern also supports a partial mediation pathway in the moderate and high self-esteem groups.

As in the White sample, depressive symptoms and BMI were positively associated with BD among South Asian participants, but self-esteem did not moderate the direct association between parental teasing and BD ($B = -0.51$, $t(143) = -.92$, $p = .362$). Therefore, parental teasing related to BD similarly across different levels of self-esteem within the South Asian sample.

## Discussion

The current exploratory, cross-sectional study examined the associations between parental weight teasing and body dissatisfaction (BD) among White and South Asian females living in Canada, as well as the potential mechanisms underlying this relationship. Overall, South Asian females reported higher rates of parental weight-related teasing compared to White participants. Additionally, the association between parental weight teasing and BD for both White and South Asian participants included a mediational role of depressive symptoms, whereby higher weight teasing was correlated with greater depressive symptoms and concurrent BD. Interestingly, our results showed that the association between teasing and depressive symptoms was moderated by self-esteem, where teasing and depressive symptoms co-occurred more strongly for those with high self-esteem (White) or high to moderate self-esteem (South Asian). Finally, self-esteem was directly associated with BD in the White group, but this pattern was not observed among South Asian participants, for whom teasing and depressive symptoms were associated with BD regardless of self-esteem level. The sections below contextualize these findings within the existing literature and identify potential mechanisms for future investigation.

Consistent with previous research [41–44], we found that South Asian participants reported higher levels of parental weight-related teasing as compared to White participants. Conversely, South Asian and White participants did not differ with respect to self-reported depressive symptoms, self-esteem, BD, or BMI. These findings contribute to the growing body of literature that challenges the notion that BD is primarily a concern among White females. Although BD has traditionally been viewed as most prevalent in this group [45], emerging research suggests that non-White females may experience similar or even greater levels of BD [46–48].

Our results further indicated that the relationship between parental weight teasing and BD was mediated by depressive symptoms, whereby increased weight teasing was associated with greater depressive symptoms and subsequently increased BD for both White and South Asian participants. This pathway underscores the critical role of mental health in understanding how parental behaviors impact body image. Notably, we also found that the relationship between teasing and depressive symptoms was moderated by self-esteem, and differences were observed between ethnic groups. For White participants, weight teasing was more strongly associated with depressive symptoms for those with high (i.e., 1 SD above the mean) compared to moderate (i.e., mean-level) and low (i.e., 1 SD below the mean) levels of self-esteem. This counterintuitive finding suggests that high self-esteem among White females may not provide a protective buffer against the negative impacts of weight teasing; instead, it may heighten the sensitivity to such teasing, possibly due to a stronger investment in body image ideals.

In contrast, among South Asian participants, weight teasing was associated with increased depressive symptoms in those with high and moderate self-esteem, but not in those with low self-esteem. These differences may reflect culturally specific meanings of self-esteem and body image. For example, in many South Asian cultures, familial approval and adherence to cultural norms are highly emphasized, and self-worth may be more tightly linked to meeting appearance-related expectations [20]. Internalization of both traditional and Western beauty ideals may also increase vulnerability to teasing and dissatisfaction [49]. Given previous evidence linking self-esteem to protective factors such as self-compassion

and body acceptance [50], the current findings raise the possibility that parental weight teasing may diminish the usual benefits of higher self-esteem. These hypotheses are speculative and warrant future research to determine whether the mechanisms driving these interactions are consistent across cultures or context-specific.

Given the stronger association between parental weight teasing and depressive symptoms among South Asian participants, the current findings may reflect distinct psycho-social pathways through which BD manifests in non-White females. Within South Asian communities, where familial influence is often highly emphasized [27], weight-related teasing by parents may be more closely linked to depressive symptoms, which in turn are associated with BD. In contrast, among White females, parental teasing may be more closely associated with self-esteem, which shows a stronger direct association with BD. Moreover, the relationship between self-esteem and BD may be more multifaceted in South Asian compared to White females, given that factors such as racial teasing, personality traits, skin color, and acculturation have been shown to moderate the presentation of BD in South Asian samples [28,51]. Further research is needed to elucidate how self-esteem functions in relation to BD among South Asian females, and to better understand the culturally specific mechanisms underlying these associations.

Finally, in our sample, self-esteem was positively associated with BD among White females, but this relationship was not observed among South Asian females. Conversely, most prior research has identified an inverse association between self-esteem and BD across cultural groups (e.g., [51]). For example, Iyer and Haslam [52] found that self-esteem predicted BSQ scores among American South Asian females, although the absence of reported mean self-esteem levels limits direct comparison with the present findings. The divergence between our results and previous work may reflect differences in baseline self-esteem across samples. Alternatively, high self-esteem may not consistently serve as a protective factor. This perspective is supported by the concept of "contingent self-esteem," in which global self-worth is heavily dependent on meeting specific appearance standards [53,54]. In such cases, higher self-esteem scores may reflect confidence that is strongly tied to body image, rather than a more general sense of self-worth. This could increase investment in achieving an idealized appearance and heighten sensitivity to weight-related teasing, thereby exacerbating BD [55]. In contrast, the absence of this pattern among South Asian females may reflect differences in the degree to which self-esteem is appearance-contingent or the influence of alternative sources of self-worth (such as community, family roles, or non-appearance-related competencies) that buffer the impact of body image concerns [20,56].

Limitations of this study should be considered. Firstly, participants did not complete an acculturation measure, which limits our ability to draw conclusions about how cultural adaptation processes may have shaped responses or moderated associations between parental teasing, depressive symptoms, and BD. The study sample was composed entirely of undergraduate psychology students, which may also limit generalizability; however, this population is known to experience elevated rates of BD, making it a relevant group for targeted research and intervention [57]. Additionally, the measures used in this study were not specifically normed for South Asian populations, which may limit their cultural validity. Differences in how key constructs such as self-esteem, depressive symptoms, or teasing are conceptualized and expressed across cultures could introduce item bias or measurement error [28,58], and future research should work toward establishing culturally sensitive assessment tools in this area.

Another important limitation concerns the cross-sectional design of the study, which restricts our ability to draw causal conclusions about the observed relationships. While our mediation model suggests that parental weight teasing contributes to depressive symptoms, which in turn influence BD, it is equally plausible that individuals experiencing greater psychological distress may be more likely to recall or interpret parental comments as teasing. This raises the possibility of reverse causation, in which existing emotional symptoms shape perceptions of past parental behavior rather than the other way around. Relatedly, parental weight-related teasing was assessed retrospectively, whereas BD, self-esteem, and depressive symptoms were measured concurrently. This temporal mismatch introduces the possibility that participants' current psychological state may have biased their recollection of teasing experienced between the ages of 5 and 16, potentially inflating associations between recalled teasing and current symptoms. Additionally, unmeasured confounding

variables (such as broader family dynamics, parental mental health, trauma history, or sociocultural stressors) may simultaneously affect both the likelihood of being teased and the development of depressive symptoms or BD, contributing to the associations observed. Compounding these challenges is the role of BMI, which we included as a covariate to adjust for baseline group differences but which may itself lie on the causal pathway between lifestyle factors (e.g., diet, physical activity) and both teasing and BD. Controlling for a potential mediator, rather than a true confounder, can introduce statistical bias and distort effect estimates, a concern well documented in the epidemiological literature (Schisterman and colleagues, 2009) [59]. Moreover, alternative causal sequences are plausible, such as higher BMI increasing the likelihood of teasing and BD, or body dissatisfaction influencing eating behaviors and subsequent changes in BMI. Without longitudinal data, we cannot determine the directionality of these relationships or rule out the possibility that teasing, depressive symptoms, BD, and BMI are interconnected in more complex or bidirectional ways.

Finally, the current study focused exclusively on young adult females. This decision was made to reduce model complexity and focus on a population historically at higher risk for BD, but it limits generalizability across genders. While previous work has found that parental weight teasing is more common among females [60], males also experience teasing-related distress and may show different patterns of association with BD and depressive symptoms. Future research should examine these variables across gender and cultural groups to better understand how body image concerns manifest in diverse populations.

In conclusion, the current study highlights the association between parental weight teasing and BD among White and South Asian females living in Canada, emphasizing the importance of considering ethnic and cultural differences in the etiology and maintenance of BD. Parental weight teasing was linked to depressive symptoms, mediating the relationship between teasing and BD across both groups. Self-esteem moderated the effects of weight teasing on depressive symptoms differently for each ethnic group, suggesting complex interactions between cultural factors, development of self-schema and individual resilience

. Given the strong cross-cultural relationship observed between parental weight teasing and BD, healthcare professionals may benefit from increased attention to familial dynamics in treatment. Psychoeducation on the negative impacts of parent-driven weight teasing on children's body image may reduce the impact of family-based factors in the development of BD. Although the findings suggest that variables associated with BD may be similar in White and South Asian samples, additional research is needed to better inform prevention and education efforts for this under-researched population.

## Supporting information

**S1 Checklist.**
(DOCX)

## Author contributions

**Conceptualization:** Dilpreet Lidder, Maya Libben.

**Data curation:** Dilpreet Lidder.

**Formal analysis:** Amané Halicki-Asakawa.

**Investigation:** Dilpreet Lidder.

**Methodology:** Dilpreet Lidder, Maya Libben.

**Resources:** Maya Libben.

**Supervision:** Amané Halicki-Asakawa, Maya Libben.

**Writing – original draft:** Amané Halicki-Asakawa, Dilpreet Lidder.

**Writing – review & editing:** Amané Halicki-Asakawa, Maya Libben.

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
