## [Decision Letter · Decision Letter 0]

28 Mar 2025

PMEN-D-25-00060

Parental Teasing and Body Dissatisfaction in Caucasian and South Asian Females: A Moderated Mediation Model

PLOS Mental Health

Dear Dr. Libben,

Thank you for submitting your manuscript to PLOS Mental Health. After careful consideration, we feel that it has merit but does not fully meet PLOS Mental Health’s publication criteria as it currently stands. Therefore, we invite you to submit a revised version of the manuscript that addresses the points raised during the review process.

We look forward to receiving your revised manuscript.

Kind regards,

Catherine Malboeuf-Hurtubise, Ph.D., psychologist

Academic Editor

PLOS Mental Health

Journal Requirements:

Reviewers' comments:

Reviewer's Responses to Questions

**Comments to the Author**

1. Does this manuscript meet PLOS Mental Health’s publication criteria ? Is the manuscript technically sound, and do the data support the conclusions? The manuscript must describe methodologically and ethically rigorous research with conclusions that are appropriately drawn based on the data presented.

Reviewer #1: Partly

2. Has the statistical analysis been performed appropriately and rigorously?

Reviewer #1: I don't know

3. Have the authors made all data underlying the findings in their manuscript fully available (please refer to the Data Availability Statement at the start of the manuscript PDF file)?

Reviewer #1: No

4. Is the manuscript presented in an intelligible fashion and written in standard English?

Reviewer #1: Yes

5. Review Comments to the Author

Reviewer #1: This could potentially be suitable for publication but needs a major revision to address limitations, particularly in relation to conclusions drawn based on cross-sectional data.

Throughout, there are examples of language that infers causality or temporal sequence - but there is no longitudinal data to support these statements.

The role of reverse causation and unmeasured confounders is ignored (differences in diet between Caucasian and South Asian could influence BMI, influencing dissatistfaction or teasing in turn). So is the role of genetic factors and possible common unmeasured causes.

You have a relatively small cross-sectional sample, and must not step beyond that data. It can add value to the literature, but is exploratory in nature, until higher quality longitudinal data can be identified.

That connects to my second concern which is around the exploratory nature of how the data were analysed. There was no pre-registration of the analysis plan (e.g. Open Science Framework or similar). There are predictors/exposures, outcomes/dependent variables, possible mediators [limited by cross-sectional data, they were measured at the same time point], and moderators / subgroups. There is no clear research question that positions the status of these clearly upfront. Sometimes self-esteem is treated as a moderator, sometimes a covariate, in the text. This gives the whole thing an exploratory feel, and it would be better in my view to position the manuscript as exploratory throughout.

The main strength is having two samples with the same measures, to compare Caucasian and South Asian females. The figure shows that parental weight teasing has a stronger 'direct' effect to body dissatisfaction for South Asians, whereas more of the association is mediated by depression in Causasian. Direct effects in reality, are unmeasured mechanisms/mediators, so in the discussion you need to reflect more on what those might be (cautiously). The figure is clear, but doesn't need BMI - all outcomes and mediators and predictors should controlled for covariates age, bmi etc. However - adjusting for covariates that might be on the causal chain can produce biased estimates (there is an established literature in epidemiology on why we shouldn't control for covariates that are mechanisms/mediators you may want to consult), so do this with caution. To my mind BMI is a possible mediator or even reverse causation - BMI due to differences in diet and physical activity could generate parental teasing and dissatisfaction. There is no reason why that direction/sequence is any less plausible than your hypothesized direction, without longitudinal data that can test this.

The models are otherwise OK but all scales need internal consistency measures (Cronbach's alpha) for this data set reporting (p.6 for example).

Examples of language throughout I think step beyond the cross-sectional data (not an exhaustive list): Onset, development, outcomes, correlates, emergence, subsequent, leads to, development of, impact of X on Y. This all has to be rewritten. Only in one paragraph in the discussion section, reflecting briefly on *possible* mechanisms for future longitudinal research, should this sort of conjecture appear.

You have cross-sectional correlations, nothing longitudinal, and unmeasured confounders that could explain some of these (plus the problem of reverse causation).

Far too usage/emphasis on "significance" - this is much less important than effect size (translate into clinical implications, % depressed etc, size of difference between groups) and confidence interval (reflecting precision and statistial power).

Please can the Figure be one set of results, that directly correspond to the research questions at the end of the revised introduction. If self-esteem is a significant moderator, split panels by self-esteem. So up to six panels (South Asian x Caucasian, low/med/high self-esteem). Draw out the key differences - noting that unmeasured mechanisms are stronger in South Asian (because bigger direct effect).

Structure of discussion section should be improved and all hypotheses/exploratory ideas limited to one paragraph: https://pubmed.ncbi.nlm.nih.gov/10231230/

OTHER COMMENTS

P4 The focus on Caucasian samples is partly due to 'outdated notions' but there is a more fundamental and wider issue that health research in general under-represented non-Cauasian volunteers, which must be acknowledged. Your study adds value because you are contributing data on a group where data is pauce. Focus on that strength.

P5 I was puzzled here because you say models linking teasing and BD may extend, but I was left thinking why would the mechanisms be different, why wouldn't we just assume they are the same for all parents/children. This is picked up later in the manuscript and needs foregrounding here to more strongly justify the departure point for your study.

P5 Why two types - weight, competence. What happened to competence later in the manuscript? This is not clear - competence is dropped from the figure (again, this adds to the exploratory feel of the paper).

Do not say 'the discrete mechanisms' - you picked two, why? And in the figure, it's clear these weren't the strongest mechanisms in South Asian groups (the direct effect means there are unmeasured mechanisms open for future research).

A problem here is you list depressive symptoms, self-esteem, and BMI as mechanisms - but self-esteem becomes a moderator later! This is very confusing for reviewers and readers. You need to more carefully position the status of your variables, pre-register these, and connect to more precise research questions. See https://academic.oup.com/ije/article-abstract/34/3/638/682327

The sentences 'This research enhances....' onwards do not belong at the end of the introduction. The precise research questions should be here, including proposed moderators / subgroups (see STROBE equator guidelines and checklist).

P7 Assuming participants were anonymised? Add for clarity. There is no information on recruitment - where are they recruited from and what was the criteria? See STROBE

P8 Do you mean assumptions of collineairty were met, or there was no evidence of multicollinearity?

P8 Why are CIs shown but no point estimate? Very odd.

P9 This section is much too complicated and you didn't clearly specify moderated mediation in your introduction ending. Keep it simple - if you hypothesis is that the mediation depends on a moderator, test whether a model with the interaction term fits significantly better than one without (significance is OK for that sort of test, but not for main analysis results). If it needs the interaction, split into two groups and report separately. Clarify what status self-esteem has - I did not understand this, so readers may not either.

Stop saying "significant" and explain in simple terms what the effect sizes mean in practice in absolute terms.

P10 The focus on self-esteem here is odd because the Figure, which typically shows the main results / primary research question, doesn't feature self-esttem at all.

P11 The sentence 'our findings provide support for an emerging...' needs rewriting, it is not clear.

P12 Here, it is very interesting to read that cultural beauty standards may be contradictory and more complex for South Asian compared to Cauasian females. For someone who doesn't know this, more detail is needed - and isn't this the departure point for your research question i.e. the motivation for the hypotheses? It should be up front in the introduction section. Why would we expect the models to differ - because of these nuances. Make this more explicit in a revision.

P13 'statistical factors' is the wrong phrase here because it may be confused with 'factors' in factor analysis.

P14 Reverse causation! Must be acknowledged - dietary factors and physical activity differences may influence BMI leading to depression, teasing etc. The teasing may not be cultural differences, but similar responses to BMI, driven by lifestyle differences. Of course it may be bidirectional, but that's the limitation of cross-sectional data - we don't know.

6. PLOS authors have the option to publish the peer review history of their article (what does this mean? ). If published, this will include your full peer review and any attached files.

**Do you want your identity to be public for this peer review?** For information about this choice, including consent withdrawal, please see our Privacy Policy .

Reviewer #1: **Yes: ** Gareth Hagger-Johnson

---

## [Decision Letter · Decision Letter 1]

6 Aug 2025

PMEN-D-25-00060R1

Parental Teasing and Body Dissatisfaction in Caucasian and South Asian Females: An Exploratory Cross-Sectional Analysis Using Moderated Mediation

PLOS Mental Health

Dear Dr. Libben,

Thank you for submitting your revised manuscript to PLOS Mental Health. As you may be aware, in order to be able to accept the manuscript, we require two independent reviews to have been obtained. I am very sorry that this was not done by the original handling editor. I have taken over the handling of your manuscript in order to secure the second reviewer report  and so I hope you understand that we will be requesting another round of revision in order to address their concerns. You can find the comments from the second reviewer below. I understand that this will be frustrating at this stage however I would like to let you know that I will be assessing any future changes made in-house in order to save time and the previous reviewer has signed off on the changes made in response to their report.

If you have any questions at all, please do not hesitate to reach out to me at kmontague-cardoso@plos.org and I will be happy to assist you. 

We look forward to receiving your revised manuscript.

Kind regards,

Karli Montague-Cardoso

Executive Editor

PLOS Mental Health

Journal Requirements:

1. Please include a complete copy of PLOS’ questionnaire on inclusivity in global research in your revised manuscript. Our policy for research in this area aims to improve transparency in the reporting of research performed outside of researchers’ own country or community. The policy applies to researchers who have travelled to a different country to conduct research, research with Indigenous populations or their lands, and research on cultural artefacts. The questionnaire can also be requested at the journal’s discretion for any other submissions, even if these conditions are not met.  Please find more information on the policy and a link to download a blank copy of the questionnaire here: https://journals.plos.org/mentalhealth/s/best-practices-in-research-reporting. Please upload a completed version of your questionnaire as Supporting Information when you resubmit your manuscript.

- https://doi.org/10.1007/s11199-020-01122-4

- https://doi.org/10.1016/j.bodyim.2010.05.004

In your revision ensure you cite all your sources (including your own works), and quote or rephrase any duplicated text outside the methods section. Further consideration is dependent on these concerns being addressed.

Additional Editor Comments (if provided):

Reviewers' comments:

Reviewer's Responses to Questions

**Comments to the Author**

1. If the authors have adequately addressed your comments raised in a previous round of review and you feel that this manuscript is now acceptable for publication, you may indicate that here to bypass the “Comments to the Author” section, enter your conflict of interest statement in the “Confidential to Editor” section, and submit your "Accept" recommendation.

Reviewer #1: All comments have been addressed

Reviewer #2: (No Response)

2. Does this manuscript meet PLOS Mental Health’s publication criteria ? Is the manuscript technically sound, and do the data support the conclusions? The manuscript must describe methodologically and ethically rigorous research with conclusions that are appropriately drawn based on the data presented.

Reviewer #1: Yes

Reviewer #2: Yes

3. Has the statistical analysis been performed appropriately and rigorously?

Reviewer #1: Yes

Reviewer #2: Yes

4. Have the authors made all data underlying the findings in their manuscript fully available (please refer to the Data Availability Statement at the start of the manuscript PDF file)?

Reviewer #1: Yes

Reviewer #2: No

5. Is the manuscript presented in an intelligible fashion and written in standard English?

Reviewer #1: Yes

Reviewer #2: Yes

6. Review Comments to the Author

Reviewer #1: This is much improved an now suitable for publication in my view.

Reviewer #2: Dear authors,

Thank you for giving me the opportunity to read this manuscript. This paper presents a moderated mediation analysis and comparison between the associations of parental teasing and body dissatisfaction in white and South Asian females. The topic is important and I commend the authors for their work. The authors have made substantial improvements of their manuscript during the last round of revision. However, there are still a few issues that need to be addressed before publication. Please find my comments below.

General

I strongly recommend replacing the word ‘Caucasian’ with ‘White’. Caucausian has some racist undertones and despite it being used in other publications, I strongly recommend not to perpetuate this (see e.g., https://www.careharder.com/blog/is-the-word-caucasian-racist).

Throughout the manuscript, you are switching between “depressive symptoms” and “depression” – I suggest using “depressive symptoms” throughout

I have some concerns about your results in relation to the variable self-esteem. When looking at your correlation table, it seems like higher self-esteem is associated with higher body dissatisfaction, higher BMI, higher parental weight-related teasing and more depressive symptoms. Please check the coding, this might also explain your unexpected results of the moderated mediations.

Introduction

“Additionally, Webb et al. (2020) reported that parental weight-related teasing during early adolescence was associated with emotional eating and increased over time” – What do you mean by "over time"? Do you mean the association became stronger in older adolescents, or are you referring to findings from a longitudinal study?

“The current study examined the cross-sectional association between parental teasing and BD South Asian minorities living in Westernized countries” IN Sout Asian minorities

Some restructuring of the last 4 paragraphs of your introduction is needed- provide evidence from other studies first before starting to introduce your own study.

The presentation of your hypotheses could be clarified- no all hypotheses are consistent with a moderated mediation framework, I suggest the following: “As such, we aimed to address the following research questions using a moderated mediation framework: (1) Do self-reported levels of parental weight related teasing differ between Caucasian and South Asian participants? (2) Is parental teasing associated with BD, and is this relationship mediated by depressive symptoms? and (3) Does self-esteem moderate associations between teasing, depressive symptoms, and BD, consistent with a moderated mediation framework?”

The associations between body dissatisfaction and weight-related teasing are introduced well in the introduction. However, there is very little literature cited that specifically addresses the relationships between weight-related teasing, depressive symptoms, and self-esteem—despite the fact that the moderated mediation models focus explicitly on these variables. Consider strengthening the introduction by citing relevant studies to justify the focus on self-esteem and depressive symptoms in the models.

Methods

“computer or laptop to complete the study online” – what about participants who completed the study on their phones?

Body dissatisfaction- “6-point Likert rating scale ranging from 1 (i.e., never) to 6 (i.e., always), with higher scores indicating more BD.” – should be HIGHER BD

“Each item is scored using a 4-point Likert rating scale ranging from 1 (i.e., strongly agree) to 4 (i.e., strongly disagree) and summed to create a total score, with a higher score indicating more positive self-esteem.” – check the coding of this variable

Variables

“BMI, calculated using self-reported height and weight (kg/m²), was considered a potential confounder or covariate.” -call it a covariate, not confounder

Results

Immigrant status has not been described in the methods section

Tests of mediation – italicize letters when reporting results

“Given the cross-sectional nature of the data, however, causal interpretations should be viewed with caution, as reverse causation remains a possibility.” – since this is a cross-sectional dataset, causal interpretations cannot be made at all

Tests of moderated mediation

I have concern about this analysis, as the sample size within self-esteem groups are extremely small. Even for exploratory purpose, these subgroups might be too small to support meaningful or reliable conclusions. I therefore suggest removing them from the manuscript.

Table 1 – make note more informative

Table 2- add description of reliability score (Cronbach’s alpha) below the table

Tables 3 and 4- combine this information into 1 table with one participant group above, the other one below the diagonal

Discussion

“This counterintuitive finding suggests that high self-esteem among Caucasian females may not provide a protective buffer against the negative impacts of weight teasing; instead, it may heighten the sensitivity to such teasing, possibly due to a stronger investment in body image ideals.” – these unexpected findings need to be discussed in more detail (after checking the self-esteem variable). Why do you think that those with higher self-esteem would be more invested in body image ideals?

Limitations- it should be noted that parental weight-related teasing was recalled retrospectively, while BD, self-esteem and depressive symptoms were assessed concurrently. This introduces the possibility that participants' current levels of BD, self-esteem or depressive symptoms may have biased their recollections of teasing experienced between the ages of 5 and 16.

7. PLOS authors have the option to publish the peer review history of their article (what does this mean? ). If published, this will include your full peer review and any attached files.

**Do you want your identity to be public for this peer review?** For information about this choice, including consent withdrawal, please see our Privacy Policy .

Reviewer #1: **Yes: ** Gareth Hagger-Johnson

Reviewer #2: **Yes: ** Fabienne E. Andres

---

## [Editor Report · Decision Letter 2]

24 Aug 2025

Parental Teasing and Body Dissatisfaction in White and South Asian Females: An Exploratory Cross-Sectional Analysis Using Moderated Mediation

PMEN-D-25-00060R2

Dear Dr. Libben,

We are pleased to inform you that your manuscript 'Parental Teasing and Body Dissatisfaction in White and South Asian Females: An Exploratory Cross-Sectional Analysis Using Moderated Mediation' has been provisionally accepted for publication in PLOS Mental Health.

Best regards,

Karli Montague-Cardoso

Staff Editor

PLOS Mental Health